# Perceived Facilitators and Barriers for Actual Arm Use during Everyday Activities in Community Dwelling Individuals with Chronic Stroke

**DOI:** 10.3390/ijerph191811707

**Published:** 2022-09-16

**Authors:** Grace J. Kim, Shir Lebovich, Debbie Rand

**Affiliations:** 1Department of Occupational Therapy, Steinhardt School of Culture, Education, and Human Development, New York University, New York, NY 10003, USA; 2Department of Occupational Therapy, School of Health Professions, Sackler Faculty of Medicine, Tel Aviv University, Tel Aviv 6997801, Israel; 3Occupational Therapy Services, Sheba Medical Center, Ramat Gan 52621, Israel

**Keywords:** actual arm use, upper extremity, stroke, real-world use, home-based rehabilitation

## Abstract

Background: Our aim was to gain a deeper understanding of perceived predictors for actual arm use during daily functional activities. Methods: Qualitative study. Semi-structured interview data collected from individuals with chronic stroke living in the community. Codebook thematic analysis used for the data analysis. Results: Six participants 5–18 years post stroke with moderate to severe UE impairment. Three domains were identified: Person, Context, and Task. Themes for the Person domain included mental (cognitive effort, lack of acceptance), behavioral (routines/habits, self-evaluation), and physical (stiffness/fatigue). Themes for the Context domain included social environment (being in public, presence, and actions of others) and time constraints (being in a hurry). Themes for the task domain included necessity to complete bilateral and unilateral tasks, and safety (increased risk of accidents). Conclusion: Actual arm use is a complex construct related to the characteristics of the person, contextual environment, and the nature of the task. Facilitators included cognitive effort, routines/habits, self-evaluation, and the perceived necessity. Barriers included in lack of acceptance, stiffness/fatigue, being in public, being in a hurry, and risk of ac-cidents. Social support was both a facilitator and a barrier. Our results support the growing call to adopt a broader biopsychosocial framework into rehabilitation delivery.

## 1. Introduction

Weakness of the upper extremity (UE) contralateral to the brain lesion is very common post stroke, and contributes to long term disability experienced by individuals adjusting back into the community [1]. UE rehabilitation is a challenging issue. At six months, 38% of patients report some level of hand dexterity, and only 11.6% achieve full recovery [2]. Individuals with limited UE function may have difficulties performing daily activities since bilateral hand use is essential to easily perform daily activities, such as getting dressed, making a salad, and washing dishes. A recent study demonstrated that hand function of the affected UE (assessed using the Box and Block Test) significantly contributed to 19.3% and 19.5% of the variance of independence in basic and instrumental activities of daily living (respectively) at six months post stroke [3].

From a motor learning perspective, the acquisition of new motor skills or the re-acquisition of lost skills occurs in stages over time. Cognitive demands and learning goals are distinct for beginners, intermediate learners, and advanced learners [4]. The specific learning characteristics at each stage can guide clinicians to provide the appropriate instruction (verbal cues, goal of task) specific to the stage of learning for their clients.

Recent evidence suggests that improvements in motor ability achieved in the clinic do not automatically generalize to actual use of the affected UE in the home and community [5,6,7,8]. This discrepancy has underscored the importance of UE interventions to target both the reduction of motor impairment and the carryover of these gains into actual arm use in real-world settings to maximize an individual’s functional independence and quality of life.

Actual arm use is defined as the spontaneous integration and use of the affected UE in real-world situations outside of the clinic [9]. The terminology used to describe this concept has not been consistent in the stroke literature. Other terminology used in the literature includes real-world use, UE performance in daily life, and daily use. The term “actual arm use” will be used in this paper. Actual arm use can be measured using self-report questionnaires such as the Actual Amount of Use Test [10], Motor Activity Log [11], and Ratings of Everyday Arm-use in the Community and Home (REACH) [12]. Motion sensors can quantify continuous UE motion for 24–72 h in real-world settings [13,14]. Individuals with stroke reported use of the arm in everyday tasks as the single most important indicator for UE recovery [15], underscoring the urgent need to develop interventions that specifically target actual arm use. In order to do this, the underlying factors that help or hinder actual arm use must be better understood.

A review of the existing literature provides some insight regarding the underlying factors associated with various aspects of UE recovery. A 12-week longitudinal study reported the following self-perceived barriers to UE recovery: not enough upper extremity movement (73%), too many other things to deal with (46%), and feeling that they cannot do things correctly (46%) [16]. A review of qualitative studies provides further understanding of the perceptions, beliefs, and experiences of individuals post stroke specific to their UE recovery. Commonly reported barriers for UE recovery included: not enough time in acute rehabilitation [17], too much focus on lower extremity compared to UE rehabilitation [17,18], sensory impairment [19,20], not enough movement in the arm [15,21], spasticity [15,22], frustration when attempting to use the arm [15,17,22], sustaining motivation to practice [17,21], and low expectations of clinical staff [17,20,23]. Factors associated with facilitating UE recovery included feedback from therapists and the ability to see small changes [17,21], positive emotional support from therapists or caregivers, adjusting tasks and actual use of the arm [15,21,22].

The majority of existing literature reviewed provides insights related to UE recovery broadly, with one study [22] focused on delineating the factors associated with hand use. The results underscored the importance of task modification (adjusting positioning, practicing components of tasks) and the adoption of detached focus to decrease frustration to facilitate hand rehabilitation [22]. However, more research is needed from the client’s perspective that can contribute to actual arm use specific to the completion of daily activities in their natural settings. To address this gap in the literature, we conducted a qualitative study with community-dwelling individuals with stroke. The aim of study is to gain a deeper understanding of the perceived facilitators and barriers associated with actual arm use of the affected UE during the completion of common daily activities. The findings may inform and guide occupational therapy practitioners in the development of home-based rehabilitation interventions to increase actual arm use.

## 2. Materials and Methods

### 2.1. Population

Participants were included if they met the following criteria: had a diagnosis of stroke, were living in the community, absence of major cognitive impairment (Mini-Mental status examination score > 24) and some level of functional difficulties using their affected upper extremity, (REACH scores > 1). Recruitment strategies included study flyers in the community and a local stroke registry. A purposive sampling method was used for the stroke registry and potential participants were contacted who would most likely meet inclusion criteria. The protocol was approved by the Institutional Review Board at New York University and all participants provided written informed consent.

### 2.2. Procedure

Data were collected as part of a separate study to establish content validity for a new outcome measure for actual arm use in community-dwelling individuals with stroke. Semi-structured interviews were conducted by the primary author (GK). Participants were asked how much and how they used their affected arm during specific basic and instrumental activities of daily living (e.g., dressing, grooming, toileting, bathing, laundry, meal preparation, medication management, phone use). Participants were also asked how confident they felt using their affected arm in different situations (e.g., feeling tired, in a rush, in public, with family or friends). See Appendix A for a copy of the semi-structured interview guide. Interviews took place in the Occupational Therapy Department research lab at New York University and lasted 45–60 min. Demographic data were collected to characterize the participants including age, time post stroke onset, affected side, household status, and cognitive status. The Fugl-Meyer Assessment (FMA) and REACH measures were used to collect data related to UE motor impairment and everyday use. FMA is a valid and reliable tool; scores range from 0 (no active movement) to 66 points (full active movement) [24]. The REACH is valid and reliable to assess perceived daily-use of the affected upper extremity outside of the clinical setting, scores (levels) range from 0 (no use/exercise only) to −5 (full use) [12].

### 2.3. Analysis

Codebook Thematic Analysis was used to analyze the interview data [25,26]. An initial codebook was created by all co-authors through independent scrutiny of the data. All co-authors independently completed open coding of participants 1 and 2, guided by the study aims and theoretical frameworks (deductive coding). Final determination of first-round codes was established based on group consensus. In the second round, all co-authors re-coded participants 1 and 2 based on agreed-upon codes. Additional second-round codes were identified directly from the data through inductive coding and added to the existing codebook. First-round codes were also modified as needed (e.g., definition of terms adjusted, child codes added). Coding was then completed on participants 3–6 using the established codebook.

### 2.4. Analyzing and Identifying Themes

Once coding was completed for all participants, coded excerpts were downloaded from Dedoose software (Version 9.0.54, SocioCulural Research Consultants, LLC., Los Angeles, CA, USA) and connected and summarized to determine dominant domains, themes, and subthemes. The identification of dominant themes and associated subthemes was an iterative process. Group discussions with all authors were completed to identify and agree on dominant themes and to further identify and organize subthemes within each dominant theme. Disagreements with themes were resolved by group discussion until consensus was achieved. Additional rounds of coding were completed by at least two authors for further analysis of subthemes within each dominant theme.

### 2.5. Interpretation of the Themes

Relationships within and between themes were determined by an explanatory framework consistent with the data. Themes and subthemes were further refined and relabeled to capture the meaning of each category.

### 2.6. Rigor

Semi-structured interviews were audiotaped and transcribed verbatim. The written transcription files were uploaded into Dedoose for coding completed by co-authors. The codebook was entered into Dedoose and modified with additional inductive codes and updated definition of codes as they occurred. The use of Dedoose facilitated consistency and automaticity of the coding process and notes entered for all three co-authors. This served as an evidence trail to increase rigor of the coding process.

### 2.7. Reflexivity

All authors are occupational therapists with clinical and research experience in stroke rehabilitation. Our interpretation of the data was informed by an occupation-focused lens including the Occupational Therapy Practice Framework-4, neurorehabilitation theoretical lens including motor learning and neuroplasticity, and health behavior change models such as the Transtheoretical Model.

## 3. Results

Six participants (five women and one man) were interviewed. The description of each of the participants appears in Table 1. Participants were 5–18 years post stroke (median 10 years), had moderate-to-severe motor impairment of their affected upper extremity (median FMA score 27.5 points out of a maximum 66 points) [27], and limited daily hand use according to the REACH (median level 1.5 out of maximum 5 levels). All participants were independent with community mobility with or without the use of a cane.

Three primary domains were identified as the first step in the analysis: Person, Context, and Task. Themes within the Person domain were defined as any aspect of the individual’s cognitive, psychosocial or motor systems associated with use of the affected arm. Themes identified included mental factors (subthemes: cognitive effort and lack of acceptance), behavioral factors (subthemes: routines and habits, self-evaluation), and physical factors (subtheme: stiffness/fatigue). The Context domain was defined as any aspect of the individual’s external environment associated with use of the affected arm. The primary theme identified was social environment (subthemes: being in public, presence of others, and actions of others), and time constraints with the subtheme of being in a hurry. The Task domain was defined as any aspect of the task itself that was associated with use of the affected arm. Themes for this domain included necessity (subtheme: bilateral and unilateral tasks) and safety characteristics (subtheme: increased risk of accidents). See Figure 1 for visual summary of themes and subthemes.

### 3.1. Person

#### 3.1.1. Mental: Cognitive Effort and Lack of Acceptance

The mental theme included factors related to cognitive or emotional characteristics of the individual. The resulting subthemes were cognitive effort and lack of acceptance. Cognitive effort related to remembering or thinking about using the affected arm more, which seemed to provide underlying motivation to keep trying to use affected arm during activities. For example, “I want to use it really, honestly, I should be using my left hand” (P4). “I do use it, yeah. But not as often as I should” (P2). “Even over the sink and wash some dish, I have to use the right hand [affected] to soap the dishes instead of everything with the left hand” (P2). Investing cognitive effort also involved increased focus and concentration on moving the affected arm during a task. P3 said, “I just need to really concentrate and using the right arm more”, and P1 said, “I try to remember to incorporate that. Yes, I try to remember”. For some participants, being aware of the affected arm resulted in realization that they cannot do certain activities, which hindered use. P1 stated “because it doesn’t work and I know it doesn’t work, so I actually forget about it”.

Some participants explained that they still have not fully accepted the fact that their upper extremity was non-functional due to the stroke. Lack of acceptance was related to high expectations for success and ultimately a barrier to actual arm use. P3 stated, “I want to be able to use it in the right way. For me, that’s what I will accept”. P5 expressed a lack of acceptance by comparing their affected arm to what their arm could do prior to the stroke: “I could pick up my guitar and play some chords and just do things that I measure my disability by involuntarily. Things that I miss from my pre-stroke days”…

#### 3.1.2. Behavioral: Routines/Habits and Self-Evaluation

The behavioral factors identified participants’ action-oriented activities that facilitated or impeded actual arm use. Participants reported having routines and habits to facilitate arm use. Routines/habits were described as consistent behavior integrated into daily activities. P2 described her routines and habits that she has created in order to increase use of her affected arm: “Every time I go to the refrigerator, I make sure I open it with the right [affected] hand” and “I would get the bread and I always like to use the right hand [affected side] to see how good I could put the, if it’s not peanut butter or spread, to spread it on the bread”. Interestingly, P2 had the highest scores on the FMA and the REACH of all participants.

Self-evaluation included observations and informal feedback on performance of the affected arm during completion of daily tasks. Participants who reassessed their current performance were able to use that information to motivate themselves and continue to keep working on improving the arm: “I’m still improving. I have a lot of work to do. I have to put in the work to get better” (P2) and “over each activity that I’ve done over the past, I’ve learned that I really can do more than I thought I could. That gives me confidence” (P3). For other participants, self-assessment of their current performance compared to their pre-stroke ability led to decreased use or limitations in use of the affected arm: “I spent a lot of time in the kitchen in my life, so I find myself measuring my abilities and advancements if there are any in that context. Preparing food and realizing that I can’t cook certain things because, I can’t use a fry pan because I can’t stabilize the arm of it with my affected hand” (P5).

#### 3.1.3. Physical: Stiffness/Fatigue

Physical factors included symptoms or sensations related to body function experienced by participants. The subtheme identified was stiffness/fatigue related to their upper extremity, that usually impeded use of the affected arm: “I know when I’m tired, when I’m…I’m more stiff” (P4) or “… It’s the tone. Especially like if it’s cold, I have to cover up myself…” (P2). P3 connected fatigue with decreased arm performance: “I’m tired, so I’m not going to do well”.

### 3.2. Context

#### 3.2.1. Social Environment: Being in Public, Presence of Others, and Actions of Others

Factors within the social theme described aspects of the social environment external to the participant, which could facilitate or impede actual arm use. Key subthemes of the social environment included being in public, the presence of others, and the action of others. Being in public impeded arm use because participants felt increased pressure to perform well. For example, P6 said, “But when you’re in public, it’s not a big deal, but you don’t want anything to go wrong. So you tend to go a little less confident”. Using the affected arm in public also increased feeling embarrassed: “I really don’t use it in public …. Well I wouldn’t try to use it. I would be embarrassed” (P3). P5 minimized using his affected arm because he did not want to strangers to feel sorry for him: “When I’m around strangers in public or something, I feel like I’m playing the stroke card if I’m showing too much. You know? I don’t want to solicit or elicit sympathy or pity or something like that”.

Using the arm in the presence of others was both comforting and stressful depending on the audience. Participants made a distinction between the presence of family and friends versus strangers. The presence of their spouse, children or close friend could be comforting. “What about when you’re with your family or friends? That’s very confident, Very. I’m not afraid or ashamed to use my hands, you know”? (P2). However, participants reported greater risk of judgment or pity from others when asked about using their affected arm the presence of strangers. P3 said, “I don’t know if they do. But, I just feel like they are looking, looking. I wouldn’t know if they’re judging or not, but looking”. P5 wanted to avoid looking needy and did not want other people to feel pity for him: “I don’t want to look like too much of a freak in the laundry room. So I downplay my... inactivity in my affected side”.

The actions of others resulted in participants receiving emotional or physical support. Some participants receiving encouragement or reminders to use their affected arm. P4 expressed that “Because I’m really not using it [affected hand], and my friend is telling me to, you know, involve it so that will improve. He’s telling me I’m neglecting it, and it’s not good”. P1 reported, “somebody taught me that it’s not just a piece of ligament just hanging. You’ve got to incorporate that piece of ligament. I try to remember to cooperate that. Yes, I try to remember”. However, participants more often expressed decreased opportunities to use the affected arm because of the physical assistance provided by someone around them. P3 reported that “most of the time my son is with me anyway. I have to fight with them sometimes to push, for them to let me push the cart”. P1 said, “Most of the time, if I need a knife and fork to cut it, it’s already done [for me]”.

#### 3.2.2. Time Constraints: Being in a Hurry

Participants reported that being in a hurry usually resulted in not using the affected arm. If participants had time constraints while completing a task, this resulted in relying more on their unaffected side to complete tasks quickly or accurately: “I’m in a hurry, I wanted to do this and that … So, I try to do everything faster by using the left hand instead of involving the right hand [affected side]” (P2). P3 reflected similar response: “If I’m in a rush, I don’t use it. It slows me down”.

### 3.3. Task

#### 3.3.1. Necessity: Bilateral and Unilateral Tasks

The need to integrate both hands in order to complete a task was a common facilitator for actual arm use. “I use it when I have to” was a common sentiment across all participants. If a task could be completed with only their less-affected arm, participants often did not use their affected arm.

P6 shared the way she does her laundry: “So getting it out of the washer…I have a cart and so then I push it with both hands. And then I don’t use it to get in and out of the washer or dryer, but then folding, I use both hands”. P1 also explained, “… washing your face, brushing your teeth. …, putting on makeup, brushing your hair. That will be the left hand [less-affected side] because only if I need to hold something I need assistance”.

P5 said, “I’ve used my left hand [affected side] to keep a door open while I’m putting clothes in or taking them out so it doesn’t slam on me. As I do with refrigerator door sometimes. Sometimes I can just, it’s just enough to keep it open, a little bit, or enough”.

P1 admitted, “when I’m home alone and I need something done. I find a way to [use] this hand assist me to do almost everything that I need doing, yes”.

Unilateral tasks that required something to be done on the less affected arm or side of the body facilitated use of the affected arm. For example, participants reported using their affected arm when the less affected side needed washing, drying, lotioning or dressing. P6 said, “So I use shower gel. And so, I put it on a stick like to do it, and so then I use my affected arm to wash my [less-affected arm], you know? So, yeah, I do that everyday”. P2 also explained “I wash my left side, I wash my body with the right hand [affected side] under the arm”.

#### 3.3.2. Safety: Increased Risk of Accidents

The safety characteristics of a task influenced actual use of the affected arm. Perceived increased risk of accidents while completing tasks such as holding a cup of hot water, cooking or holding something heavy were a clear barrier for using the affected arm. Tasks that could increase the chance of accidents (something spilling or falling) or increase risk of physical harm to the participant decreased use of the affected arm.

P2 said, “…I make sure I put it in my left hand [less-affected side] to throw the detergent because I don’t want it to spill all over the place”. P5 explained how he functions in the kitchen: “Or use a pancake flipper with the affected hand while I’m stabilizing the pan with the right hand with the unaffected head. It just gets too dangerous…Yeah, safety does come into play in the kitchen”. P6 reiterated this by explaining “because I wouldn’t put anything in my affected hand where they could drop and it could really be bad”.

### 3.4. Facilitators and Barriers

As a subsequent step, we organized themes and subthemes into facilitators and barriers to gain additional understanding of the results (Figure 2). Themes and subthemes that were reported by two or more participants were included. Facilitators and barriers included themes from all three domains, which are color-coded in Figure 2. In the next section, we discuss in detail each category and the implications for actual arm use.

## 4. Discussion

The underlying factors for integration of the affected arm in everyday activities expressed by individuals with chronic stroke living in the community were summarized and synthesized. These factors were organized into three overarching domains: Person, Context, and Task. We further synthesized each of the themes and their subthemes into (1) facilitators and (2) barriers. Some themes were both facilitators and barriers. We further discuss these categories to gain a deeper understanding about actual arm use (Figure 2).

### 4.1. Facilitators

In the Person domain, the factors that facilitated actual arm use were cognitive effort (mental theme), and routines/habits and self-evaluation (behavior theme). Our findings are supported by previous literature, which has reported strong correlations between cognitive effort and motor coordination and execution of UE motor tasks in the lab [28,29] and stakeholder perspectives [22]. For participants who do not automatically remember to use their affected arm to complete tasks, “thinking about” or “remembering” to use the affected arm may be a helpful step before “doing” the task. This multi-step approach is supported by theories of skill acquisition within the motor learning literature. Fitts and Posner’s Three Stage Theory posits that the acquisition of new skills or the re-acquisition of lost skills occurs over three stages: cognitive, associative, and autonomous stages. The initial phase requires many cognitive resources because the learner must figure out the best way to complete a desired task [4].

Self-evaluation was consistently a facilitator for actual arm use. The ability to see small changes in the affected UE (i.e., performance feedback) observed by the self or a therapist is motivating and important for clients to keep training [17,21]. Self-evaluation, or the ability to identify mistakes and self-correct, is characteristic of the associative stage of motor skill acquisition, which uses fewer cognitive resources and engages in problem solving by adapting and completing the task under different environmental conditions [4].

Participants who created a habit to use their affected arm during certain tasks (e.g., keeping the refrigerator door open while taking out food) consistently reported use of their arm for those tasks. Habituation is defined as a semi-autonomous process of organizing behavior into routines or repeated patterns without conscious effort [30]. Within the skill acquisition framework, the use of habits and routines to complete tasks is characteristic of the autonomous phase, which is the last stage of skill acquisition [4]. Within a health behavior framework, creating habits minimizes cognitive effort and decision making, which helps individuals increase a desired action or behavior [31]. Habit formation facilitates exercise and physical activity in stroke, cardiac rehabilitation, and general adult populations [32,33,34]. Interestingly, facilitators within the Person domain encompassed all phases of skill acquisition, suggesting that the participant’s stage of motor reacquisition must be taken into consideration when addressing actual arm use in everyday activities.

These findings also highlight the role of self-efficacy on actual arm use. In a recent study [35], self-efficacy was found to be correlated with perceived daily hand-use in 22 individuals with high functional capacity. UE self-efficacy was task or situation-specific, which is supported by our results. Interestingly, this suggests that self-efficacy for actual arm use is task specific, which is a relatively new concept, and should be further investigated.

Within the Task domain, the necessity to complete tasks bilaterally was the most common perceived facilitator for actual arm use. Data from UE wearable sensors have demonstrated that adults without neurological conditions typically integrate both sides to complete most everyday functional tasks [14,36]. It is our natural inclination to spontaneously use our UEs bilaterally to complete tasks in the real world. Activities such as washing hands, holding a large object, and pushing a cart illustrate the advantage of bilateral arm use. The use of the affected arm for performing unilateral tasks can be limited depending on the functional level [37]; however, participants used their affected arm to help or assist during bilateral tasks or tasks that should be done for the less affected side, if they perceived a need for it.

### 4.2. Barriers

The lack of acceptance of the individual’s current ability (Person domain) was consistently reported to be a barrier for actual arm use. Existing literature on coping after stroke suggests that while acceptance increases over time, coping after stroke is an ongoing and dynamic process because of continued changes in their body, mental and emotional states [38]. Some individuals respond by comparing their current situation to their life before the stroke, often resulting in frustration at their slow progress [38,39]. Acceptance of a new condition and the formation of a new self-identity are associated with increased participation [40] and higher life satisfaction [38,39].

Not surprising, stiffness/fatigue (Person domain) was a clear barrier for arm use. Stiffness or spasticity can be triggered by cold weather, pain or physical effort. Our results are supported by existing literature reporting the negative relationship that spasticity has with passive range of motion of the wrist and elbow [41], motor recovery beyond 3 months [42] and UE recovery [15,22].

Interestingly, motor impairment was not a significant barrier for actual arm use in our results. This was surprising since arm movement has been identified as a key factor contributing to upper limb recovery [14] and actual arm use [37]. An explanation for this could be that participants were not asked to identify facilitators and barriers for arm recovery but were asked to describe how they performed specific daily tasks with their affected upper extremity after they rated the readiness to use the affected arm in daily tasks. This shift in perspective may have influenced participant responses because they may have assumed their motor impairment was already acknowledged.

Being in public (Context domain) was a consistent barrier to arm use. Participants associated feelings of embarrassment, less confidence, and increased risk of exposure when they talked about affected arm use in public. The potential negative reactions of strangers also decreased the likelihood of participants using their affected arm in public. The implicit pressure to minimize or not bring attention to their disability in public suggests a society bias toward individuals with disability [43]. Our results support existing literature [18] indicating that some societal factors may have a negative impact on individuals with stroke and their recovery process. This area should be investigated further and would require the use of a critical disability framework that conceptualizes disability within the context of the larger society [44,45].

Participants were hesitant to use their affected arm when they were in a hurry (Context domain). The time constraints expressed by participants were self-imposed but in response to external situation. They preferred to use their less-affected arm to text someone back quickly or if they were in a rush to be somewhere or get something done. In all cases, they chose to use their less-affected arm to complete the task more quickly and efficiently.

Participants consistently reported hesitancy to use the arm if the characteristics of the task increased the risk of spills or accidents (Task domain). One explanation for this could be that real-world hand tasks such as carrying a cup of water to the table or pouring laundry detergent into the washer increase the motor and attentional demands of the task. The literature on dual tasks demonstrates that increased motor and cognitive demands decrease UE performance time and motor control in individuals with stroke compared to healthy populations [46,47], so it is not surprising that tasks completed at home with increased hazards and decreased safety would be a deterrent to use the affected side.

The ratings on the REACH, which measures perceived daily hand use, are another indicator that risk or safety issues affected actual arm use. Scores on the REACH distinguished between “full daily use” (level 5) and “everyday use unless potential negative consequences” (level 4). At level 4, the affected hand is used for all daily activities except for activities that involve a risk such as shaving or carrying something hot.

### 4.3. Facilitator and Barrier

The presence and actions of others (Context domain), were both positive and negative for actual arm use. The literature on the role of social support in stroke rehabilitation supports this mixed result [48,49,50]. Previous research indicates that clinicians and caregivers can be key sources of emotional support that greatly benefit UE recovery [17,21,23], as reflected by our participants. Conversely, low expectations of clinicians can negatively impact UE recovery by limiting possibilities and decreasing hope [17,20,23]. Family members may provide too much help, eliminating opportunities for the client to use the affected side as much as possible, which was reflected by our participants. Level of function, phase of recovery, and patient’s cultural/ethnic background may also be factors to consider when addressing social support [50].

To our knowledge, this is one of the first studies to identify factors specific to actual arm use for the completion of everyday tasks in community dwelling individuals with chronic stroke. Our findings indicate that actual arm use is a complex construct related to the characteristics of the person, the contextual environment, and the nature of the task itself. These preliminary findings highlight that therapy aimed at increasing arm use at home and in the community should not only focus on the reduction of motor impairment, but take into consideration the interaction of motor, psychological, behavioral, and social factors that contribute to health and recovery [51].

### 4.4. Limitations

Interview data were collected while completing cognitive interviews for a separate project to determine content validity of a UE outcome measure, which may have influenced participant responses. Our participants were in the chronic phase of recovery and had moderate-o-severe UE motor impairment, and therefore, these results are specific to individuals with significant UE motor impairment. Our sample included six participants based on sampling needs for the content validity project and was not based on data saturation, and therefore cannot be generalized to the larger stroke population. Future studies should expand on this initial work and explore factors for actual arm use for individuals in the acute and subacute phases of recovery, and those with mild UE motor impairment.

## 5. Conclusions

This preliminary qualitative study revealed that actual arm use at the chronic stage post stroke is a multi-dimensional construct. Facilitators included cognitive effort, routines/habit, self-evaluation, and the perceived necessity. Barriers included lack of acceptance, stiffness and fatigue, being in public, being in a hurry, and risk of accidents. Social support was both a facilitator and a barrier. Our results support the growing call to adopt a broader biopsychosocial framework into rehabilitation delivery.

## Figures and Tables

**Figure 1 ijerph-19-11707-f001:**
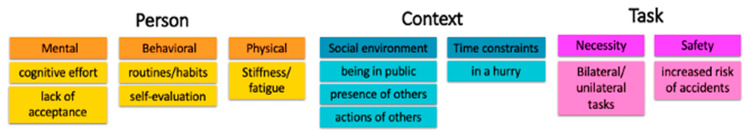
Summary of themes and subthemes based of Person, Context, and Task domains.

**Figure 2 ijerph-19-11707-f002:**
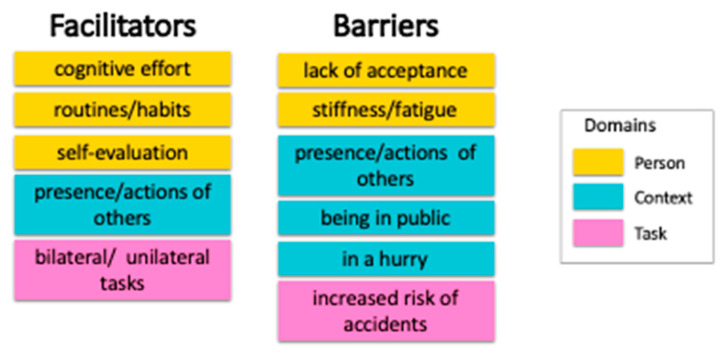
Facilitators and barriers for actual arm use based on the themes and subthemes.

**Table 1 ijerph-19-11707-t001:** Participant demographics.

ID	Sex	Age	Household	Ethnicity	Years Since Stroke	Affected Side (Dominant?)	MMSE	FMA	REACH
1	F	66	lives alone	Black	15.1	right (Y)	27	22	1
2	F	64	lives with sisters	Black	5.2	right (Y)	29	43	3
3	F	66	lives with son	Black	9.8	right (Y)	30	25	2
4	F	50	lives with spouse	Asian	9.2	left (N)	30	30	1
5	M	63	lives with spouse	White	10.3	left (N)	30	6	1
6	F	46	lives alone	South Asian	18	right (Y)	30	36	2

Note: MMSE: Mini-mental Status Exam (score range: 0–30); FMA: Fugl-Meyer Assessment, upper extremity subtest (score range 0–66); REACH: Ratings of Everyday Arm-use in the Community and Home (score range 0–5). Y = yes, N = no.

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
