# Peer review of "Perceived Facilitators and Barriers for Actual Arm Use during Everyday Activities in Community Dwelling Individuals with Chronic Stroke"

_ijerph, 2022, doi:10.3390/ijerph191811707_

Round 1
Reviewer 1 Report
This was a qualitative study examining patient perspectives of actual arm use in daily life after stroke. The authors interviewed 6 individuals who had experienced a stroke with mild to moderate upper limb paresis. The authors explore barriers and facilitators to arm use within three themes: person, context, and task. The findings are both novel and consistent with prior research that suggests actual arm use in daily life after stroke is influenced by many complex factors. This paper contributes novel information to the stroke rehabilitation community and I have a few brief suggestions to consider.
1. There have been prior critiques of referring to motion sensors or wearable devices as objective measures of actual arm use. It may be worthwhile to consider using quantitative to describe wearables vs. objective.
2. Given that the beginning of the discussion includes several mentions of the Fitts and Posner’s Three Stage Theory or the skill acquisition framework, it may be helpful to the reader to offer an initial introduction of this theory in the introduction to help lay a foundation for its mention in the Discussion.
1. It is interesting that under the person domain, there was no mention of motor function or actual arm impairment. Did this not come up in the physical theme? If not, this is a very interesting finding given that many rehabilitation professionals link actual arm use to motor impairment. Here, participants seemed to rarely mention motor impairment and instead offered a vast array of other insightful comments. It would be interesting to expand on this finding in the Discussion.
1. It appears as if Figure 2 is presenting results. Please consider moving this to the Results section.
Author Response
We thank you for your thoughtful comments, we believe that the paper has been much improved. Please see attached for itemized responses.

Reviewer 2 Report
This qualitative study aims to identified the barriers and facilitators to the incorporation of the more paretic arm in daily activities. This paper is well written and addresses an important gap related to the transfer of skills to real-life activities. This work will help guide clinicians on factors, beyond motor capacity, to target in therapy to facilitate actual arm use during every day activities. There are a few elements to address (mostly minor) before this manuscript is suitable for publication. My detailed feedback can be found below:
p.2, population: Provide a rationale for including only 6 participants.
p.2, population: In the inclusion criteria, specify the Upper extremity Fugl-Meyer Assessment ranges of scores targeted, as this will be relevant later in the description of the study population.
p.3, procedures: Provide details on the questionnaire used for the semi-structured interview. What were the topics of the questions. Consider adding a copy of the interview guide as Supplemental material.
p.3, analysis: Specify which theoretical frameworks were used to guide the qualitative data analysis. Was the analysis done independently by 2 researchers?
p.3, Analyzing and identifying themes: It is unclear when themes were generates. After participants 1 and 2, first, and then refined with the other participants? Please clarify.
p.4, results: Provide a reference for the classification used for the Fugl-Meyer. Also, there are inconsistencies in the terminology throughout the manuscript (moderate vs. mod-severe motor impairments).
p.5, behavioral: routine/habits and self-evaluation: It would be interesting in the discussion to the place the results related to the lack of hand function in relation to the literature on a potential minimal level of function to be able to incorporate the paretic arm in daily activity (see Ballaster et al., 2022 https://doi.org/10.3389/fneur.2022.804211 for example).
p.5, behavioral: routine/habits and self-evaluation: Your results highlight the role of self-efficacy on paretic arm use, but this is not mentioned here or in the discussion. Consider making links with the impact of self-efficacy on arm use here and in the discussion (p.9).
p.7, safety: I feel that both safety and activities that would have a negative consequence if not successful are described here.
p.8, facilitators: It should be acknowledged that learned/acquired non-use goes beyond memory (see Taub, E., Uswatte, G., Mark, V. W., & Morris, D. M. (2006). The learned nonuse phenomenon: implications for rehabilitation. Europa medicophysica, 42(3), 241.)
p.9, limitations: The context of the larger study should be disclosed earlier when describing the study procedures.
Author Response
We thank you for your thoughtful comments. Our paper has been improved after the revisions. See attached for itemized responses.
